# Effectiveness of Respite Care via Short-Stay Services to Support Sleep in Family Caregivers

**DOI:** 10.3390/ijerph17072428

**Published:** 2020-04-02

**Authors:** Shihomi Sakurai, Yumiko Kohno

**Affiliations:** 1Department of Nursing, Ishikawa Prefectural Nursing University, Ishikawa 9292120, Japan; 2Department of Nursing, Toyama Prefectural University, Toyama 9300975, Japan; kono-y@pu-toyama.ac.jp

**Keywords:** sleep, respite, family caregivers, actigraphy, heart rate variability

## Abstract

Family caregivers of older people who need care often experience sleep disorders, which can lead to various health problems. Although respite care is used in many countries, its effectiveness has not been fully demonstrated. We analyzed the sleep of family caregivers using actigraphy and heart rate spectral analysis to clarify changes in their sleep characteristics during short-stay respite care. Participants were all family caregivers living with an older person needing long-term care. The outcomes consisted of questionnaire responses, sleep/wake records, and R-wave to R-wave interval records. Quantitative evaluation of sleep revealed that caregivers’ median sleep time was 378.0 min, and median sleep efficiency was 94.7%. The low frequency (LF)/high frequency (HF) value was 1.722 for total sleep and 1.822 for the first half of the sleep period. The LF/HF for the first half of the sleep period was significantly different between caregiving and respite days. The respite day LF/HF was 1.567, which was significantly lower than on caregiving days. On respite days, cardiac sympathetic nervous activity among family caregivers was reduced during the first half of the sleep period. This suggests that regular use of short-stay services can improve caregivers’ sleep status, making this an effective form of respite care.

## 1. Introduction

The number of people with dementia worldwide was approximately 50 million in 2018 and is expected to increase to approximately 82 million by 2030 [1]. In Japan, it has been estimated that the number of people with dementia will reach 7 million by 2025 [2]. It is customary in Japan for adult children to take care of their parents, and the nearly 60% of recipients of long-term nursing care live with one or more family members [3].

It has been reported that family caregivers may suffer from various health problems [3,4]. In particular, two-thirds of family caregivers for people with dementia have experienced a sleep disorder [5]. Sleep disturbances increase the risk of obesity, diabetes, and cardiovascular diseases [6,7,8,9]. Features of sleep disorders among these caregivers include shortened sleep time, decreased sleep efficiency, wake after sleep onset (WASO), and less slow-wave sleep [10,11,12,13]. Nocturnal awakening is associated with the activation of sympathetic nerve activity [9]. Previously, we reported that the caregivers of people with ambulatory dementia were more likely to experience excessive sympathetic nerve activity during sleep, in comparison with other individuals in the same age range, with particular impairment to relaxation during the first half of the time asleep [14]. Previous studies investigating sleep and health among caregivers of people with dementia have also suggested that sleep disorders are associated with an increased risk of cardiovascular disease [7,8,9]. The health of family caregivers for people with dementia may therefore be negatively affected by impaired sleep. Belenky et al. [15] demonstrated enhanced daytime sleepiness and increased instances of tripping when the total sleep time was limited to ≤ 5 h. Furthermore, they reported that restoring sleep time improved sleepiness and the number of instances of tripping from the first day, with patterns returning to normal on the third day.

Respite care is used in many countries around the world as a strategy to support people with dementia [16,17,18]. However, few studies [19,20] have verified its efficacy using objective indices.

Relaxation [21], massage [22], and short-stay services [19] have been evaluated as forms of respite care that support the sleep of caregivers. The term “short-stay services” describes a type of respite care in which the care recipient is temporarily cared for overnight or for several nights by someone other than their family caregiver, to allow the family caregiver time to rest. It is recommended that individuals providing continuous care at home to a patient with dementia make use of this nursing care service. This study focused on respite care using short-stay services, to examine its effects on sleep support for family caregivers. A previous study on sleep during the short-stay respite care period used actigraphy to compare participants’ sleep during periods of caregiving and respite care; the study findings showed that caregivers who normally experienced WASO slept for longer periods during the respite period [19]. Another study among caregivers of people with dementia also found that their sleep time was longer during the respite period than while providing care [20]. These two studies indicate that respite care results in quantitative improvement of sleep among caregivers; however, the quality of sleep was not objectively evaluated.

In this study, we hypothesized that respite care using short-stay services would relieve caregivers of worry and anxiety related to caregiving, reducing tension during sleep and increasing sleep quality. We analyzed the sleep of family caregivers using actigraphy and heart rate spectral analysis to clarify changes in participants’ sleep characteristics with the use of short-stay respite care.

## 2. Materials and Methods

### 2.1. Participants

Participants were all family caregivers (hereafter, “caregivers”) living with an older person needing long-term care. Caregivers regularly used respite care with short-stay services. The selection criteria were: (1) individuals providing nursing care for people with dementia or individuals providing nighttime care, and (2) men or menopausal women, to minimize the effects of female hormones on sleep. We calculated the sample sizes using low frequency (LF)/high frequency (HF) during the sleep of caregivers of patients with dementia and caregivers from a previous study [15] as a reference. The required sample size was 11 when the detection rate was set to 80% and the significance level was set to 0.05.

Participants were recruited by providing a verbal explanation with supporting documentation to the staff of 27 home caregiving support centers in two cities and two towns in Prefecture I and requesting their cooperation in recruiting participants. We requested the participation of four institutions in the recruitment of participants. A total of 20 caregivers met the participation criteria. The staff distributed the study outline and application forms to 20 family caregivers together with stamped, self-addressed envelopes. Intention to participate in the study was confirmed by returning the application form.

### 2.2. Procedures

This study was carried out between 1 July 2015 and 28 February 2017. Researchers contacted potential participants who had returned the application form by telephone to confirm their intention to participate. Arrangements were then made to confirm a date and time for a visit. The conditions for the study nights were: (1) participants had no plans to attend any particular event, (2) the person with dementia should be at home on the night of the caregiving day, and (3) the respite day should be a night on which the caregiver is receiving respite care with short-stay services. Considering the physical and mental stress caused by caregiving, the survey of respite care was set to start on the second day after initiation, excluding the last night of the respite care period. Considering the effect of seasons on sleep, the interval between respite and caregiving days was set at a maximum of 1 month. To minimize the effect on sleep of wearing the testing equipment, participants were divided into two groups based on the order in which applications were received, with one group starting with the caregiving day and the other group starting with the respite day.

One week before the first test day, a self-administered questionnaire was sent to all participants. The participants were visited at home four times during the study. Study data were collected using the questionnaire. A heart rate sensor (WHS-1; Union Tool Co., Tokyo, Japan) and actigraph (Micro Mini; Ambulatory Monitoring, Inc., Ardsley, NY, USA) were lent to each participant, and a sleep journal was provided, in which participants were asked to write on the day following the study night.

Before going to sleep, each participant fixed the heart rate sensor to the specified site, switched it on, and checked whether the light was flashing to indicate proper functioning, and then fitted the actigraph onto the wrist of their non-dominant hand. The following morning, each participant removed the heart rate sensor and the actigraph. The researchers explained to participants that they could perform household chores and caregiving as normal and did not need to worry about the actigraph getting wet. Participants were informed that they could remove the heart rate sensor and actigraph if the devices were interfering with their sleep.

The researchers visited participants’ homes the next day to collect the testing equipment and sleep journals. Participants were asked whether they had slept as usual the previous night. The study was limited to one night of caregiving and one night of respite care, to reduce the burden on caregivers.

### 2.3. Outcome Measures

#### 2.3.1. Sleep/Waking

Activity levels were recorded using an actigraph and analyzed with AM2 software (Ambulatory Monitoring, Inc., Ardsley, NY, USA) using the zero-crossing mode [23] with 1-minute intervals. Sleep journals were used as the supporting data in the sleep analysis. The sleep variables, determined with the actigraph, were:Bedtime;Wake-up time;Sleep time: total duration of sleep in minutes (i.e., time spent in sleep between sleep onset and wake time);Sleep efficiency: (sleep time/time in bed) × 100;Sleep latency: time in minutes between going to bed and sleep onset;Amount of time scored as awake between sleep onset and wake time (WASO).

#### 2.3.2. Autonomic Nervous System Activity

A spectral analysis of heart rate variability was used as a non-invasive assessment to quantify and evaluate autonomic nervous activity. Heart rate sensors were used to record the R-wave to R-wave (R–R) intervals of heart activity during sleep. The recorded data were analyzed using the dedicated software viewer. The power spectrum was quantified into high and low frequencies using standardized frequency ranges, where low frequency (LF) was 0.04–0.15 Hz and high frequency (HF) was 0.15–0.4 Hz. HF ranges, shown as mean amplitude, were calculated as the square root of (2 × HF). Heart rate variability was evaluated every 5 min. In this study, LF/HF was used as an index for cardiac sympathetic nervous activity balance. A higher LF/HF value was considered to show increased cardiac sympathetic nervous activity.

Sleep time was defined as the period of time from first entering sleep until waking up in the morning. Autonomic nervous activity was assessed across the total sleep time and for both the first and second half of sleep time. The first and second halves of sleep time were divided by the median sleep period.

#### 2.3.3. Questionnaires

The basic participant data included caregiver demographic information, occupation, number of cohabitants, height, weight, lifestyle habits, health status, Pittsburgh Sleep Quality Index (PSQI) score, and caregiving period. We also investigated patient attributes, their association with caregiving, patients’ diseases, the Barthel Index, and Dementia Behavior Disturbance Scale (DBDS) score.

The PSQI was developed in 1988 by Buysse et al. [24] to screen sleep disorders. Doi et al. [25] developed a Japanese version of this self-administered questionnaire in 1998. The PSQI contains 19 questions, plus five more questions for individuals sleeping together in the same bed or room. The 19 questions are summarized as seven component scores, and participants are assessed over four levels, with scores from 0 to 3 points. If the total score (0–21 points) for each component is ≥ 5.5 points, the participant is judged to have a sleep disorder.

The Barthel Index is a scale for evaluating independence in daily living. Its reliability has been confirmed by Mahoney et al. [26]. Scores ranging from 0 to 10 points are distributed across 10 items relating to activities of daily living, and participants are evaluated according to their total score. Being totally independent in activities of daily living equals a score of 100 points.

The DBDS [27] is a scale for evaluating behavioral disorders and psychiatric symptoms accompanying dementia. The scale contains 28 items and is evaluated across five levels, depending on the frequency of appearance of each item (never: 0, hardly ever: 1, sometimes: 2, often: 3, and very often: 4). A higher total score indicates a greater frequency of behavioral disorders and psychiatric symptoms.

### 2.4. Ethical Considerations

This study conformed to the provisions of the 1995 Declaration of Helsinki (as revised in Edinburgh in 2000). The study was approved by the Institutional Review Board of Kanazawa Medical University (Approval Nos. 250 and 263) and the Institutional Review Board of Ishikawa Prefectural Nursing University (Approval No. 266).

Participants were provided with a written explanation, including an outline of the study. They were assured that there were no disadvantages in choosing not to participate in the study, that they could subsequently withdraw their consent, and that due consideration would be given to protecting their privacy. Participants’ consent was obtained in writing. The study required that participants wear testing equipment, so their intention to participate was confirmed three times (upon return of the initial questionnaire by mail, upon telephone confirmation of a date and time for the visit, and at the initial study visit), to ensure that participants had adequate opportunity to refuse participation if they wished.

When handling data, application forms containing personal information were allocated ID numbers and stored in a lock-fast locker, separate from the study data. The questionnaires and study data relating to caregiving days and short-stay respite care days were compared using ID numbers.

### 2.5. Statistical Analysis

SPSS version 17.0J (SPSS, Chicago, IL, USA) was used for the statistical analysis. A paired Wilcox signed-rank test was used to compare sleep status on caregiving and respite days. The level of statistical significance was set at < 5%. Measurement values are shown as median (25–75th percentile) unless otherwise specified.

## 3. Results

In total, 17 people participated in the study. Of these, 10 participants were included in the analysis. Seven participants were excluded for the following reason: following the first survey, three participants stopped the second survey, because the care recipient entered a special nursing home for old people. Two participants took off their heart rate sensors during sleep. Two participants responded that they had an unplanned visit on the next day of survey and “did not sleep as usual.”

### 3.1. Participant Characteristics

The mean age of participants was 65.05 ± 9.7 (range: 48–83) years, with four men and six women. Two participants had two cohabitants and eight had three or more cohabitants. Five participants were employed. Six participants responded that their health status was fair, three reported that they were unsure, and one did not respond. One participant had a body mass index of >25. One participant had hypertension and was taking a calcium antagonist; one participant was taking a sleep inducer. On both test days, participants were instructed to take their medication as usual. No participants were taking tranquilizing agents or female hormone supplements. The mean PSQI among participants was 5.8 ± 1.3, with six participants having scores of ≥5.5.

The care status shown in Table 1. Three participants were caring for their spouse (30%), five were caring for a parent (50%), and two cared for their spouse’s parent (20%). The average length of time for which the caregiver had been providing care was 5.0 ± 2.9 years. Four participants were providing care at night. The average patient age was 83.2 ± 11.8 years. The average Barthel Index was 46.0 ± 29.3. Seven care recipients had been diagnosed with dementia, and the mean DBDS was 24.4 ± 12.6. Three participants answered “never” when asked whether their care recipient awoke at night for no particular reason, and six answered “never” when asked whether their care recipient with dementia walked around the house at night. Two participants answered “never” to both of these questions.

### 3.2. Sleep Status on Caregiving Day

The sleep status shown in Table 2. Among caregivers, the median bedtime was 0:17 (23:12–1:11) and the median wake-up time was 6:06 (5:50–6:57). The quantitative evaluation of sleep revealed that the median sleep time was 388.0 (346.0–453.5) min, and median sleep efficiency was 96.1 (91.6–99.6)%. Median values for sleep latency and WASO were 12.0 (6.5–21.0) min and 16.5 (1.75–32.8) min, respectively. The LF/HF value was 1.722 for total sleep and 1.822 for the first half of the sleep period.

### 3.3. Comparison of Sleep on Caregiving Days and Respite Days

No significant differences were noted between caregiving and respite days in terms of bedtime, wake-up time, sleep time, sleep efficiency, sleep latency, or nocturnal awakening time. The respite day LF/HF was 1.392 for the first half of sleep, which was significantly lower than on caregiving days (*p* = 0.013). Upon comparing the differences between caregiving and respite days between those who worked and those who did not, the group that was working showed a greater difference than the group that did not work (0.790 vs. 0.210, *p* = 0.032). The LF/HF value for the second half of sleep on respite days was approximately the same as on caregiving days. No significant differences were noted for the total sleep period, but the LF/HF values were lower on respite days. No significant differences were noted between HFs on respite and caregiving days.

## 4. Discussion

In this study, we compared sleep/waking records and family caregiver sleep status on one caregiving night and one respite night, as an index for autonomic nervous sleep activity. The results demonstrated that on respite nights, the cardiac sympathetic nervous activity of caregivers was lower during the first half of the sleep period.

### 4.1. Participant Characteristics

Compared with the characteristics of family caregivers in Japan as a whole [3], our participants included a higher proportion of people providing care for their older parents. The National Livelihood Survey [3] has reported that 43% of caregivers in Japan are the spouse of the care recipient; the remainder are people caring for their parents (37%) or parents-in-law (17%). In this study, 75% of participants were caring for parents. In total, 30% of older people in Japan live with their children, but, in this study, approximately 70% of care recipients were living with their children. The high percentage of two-generational households might at least partly explain the higher proportion of adult children providing care in this study.

### 4.2. Caregiver Sleep Characteristics

Previous studies [28,29,30,31] have reported PSQI values of ≥ 6.1 for caregivers of people with dementia. The PSQI of caregivers in our study was lower at 5.8. The sleep time of participants in our study ranged from 5.9 to 8.1 hours, similar to that reported in previous studies on sleep time of caregivers for people with dementia [29,30,31,32,33]. The sleep efficiency of our study participants was 94.7%, which was higher than the <90% reported in previous studies [29,30,31,32,33]. The investigation of sympathetic nervous activity during sleep revealed that the LF/HF value for the first half of the sleep period on the caregiving day was near the upper limit of standard range [34], indicating that sympathetic nervous activity was dominant even though participants were resting. This result was similar to previous reports of LF/HF during sleep of family caregivers for people with ambulatory dementia [14].

These findings indicate that our study participants had relatively good sleep efficiency. However, sympathetic nervous activity was increased during sleep and PSQI scores exceeded those of the sleep disorder screening criteria, suggesting that caregivers’ sleep was not of very high quality.

### 4.3. Comparison of Caregiver Sleep on Caregiving and Respite Days

We found no significant differences between sleep time, sleep efficiency, sleep latency, or WASO time on caregiving and respite days. A similar study using actigraphy found that the caregivers of people with ambulatory dementia had prolonged sleep time and decreased WASO during the respite period [20]. In our study, the sleep efficiency of participants on caregiving days was 94.7%, suggesting that most participants did not experience either difficulty getting to sleep or problematic WASO. Over half of our participants were employed, and 80% had a cohabitant other than the person with dementia. These factors may explain why there were no marked changes in caregiver sleep or waking times, even when the person with dementia was not at home.

The investigation of autonomic nervous activity during sleep revealed that cardiac sympathetic nervous activity in the first half of the sleep period on respite days was weaker than on caregiving days. The values were similar to the LF/HF of 1.41 for the first half of the sleep period in non-caregivers reported in our previous study [14]. This suggests that on respite days, caregivers were under similar levels of stress as non-caregivers. A total 80% of participants responded that the person they were caring for woke during the night for no reason or walked around the house, and 40% were providing care at night. This suggested that participants were worried about their care recipient’s behavior and health, even at night. Relieving caregivers of the need to provide care with the use of respite care reduced their nighttime worries about caregiving, thereby reducing their sympathetic nervous activity during the first half of the sleep period.

Healthy adults show diminished sympathetic nervous activity upon entering sleep, with parasympathetic nervous activity becoming dominant as sleep deepens [35]. As the individual research stages 3 and 4 of non-rapid eye movement sleep, the secretion of growth hormones is promoted [36,37]. Growth hormones promote cell proliferation and adjust metabolism, helping to maintain good health. It is known that sympathetic and parasympathetic nerve activity are antagonistic and only one is activated at a time. The high levels of sympathetic nervous activity during the first half of our participants’ sleep period may have interfered with the deep sleep that usually occurs directly upon entering sleep. No differences in parasympathetic nervous activity were observed on caregiving days and respite days in our study, but sympathetic nervous activity on respite days reached the standard levels that occur during rest. This shows that the balance between sympathetic and parasympathetic nervous activity has been improved.

The triggering of sympathetic nervous activity is associated with elevated blood pressure. One large-scale study [38] suggested that individuals whose nocturnal blood pressure did not decrease were at higher risk of cerebral and cardiovascular diseases than those with decreased nocturnal blood pressure. The excessive increase in sympathetic nervous activity during the first half of the sleep period noted among our participants on caregiving days may have interfered with the decrease in nocturnal blood pressure. On respite days, sympathetic nervous activity among caregivers was approximately at the same level as in non-caregivers [14]. The regular use of respite care may therefore reduce the risk of cerebral and cardiovascular disease onset.

Our findings suggest that respite care using short-stay services can decrease sympathetic nervous activity during the first half of the sleep period, resulting in qualitative sleep improvement. Regular use of short-stay services for respite care may therefore support the health of family caregivers. This factor is related to maintaining quality of life for the care recipient and care provider, thereby enabling the continuation of care at home. Our study findings suggest that home care team members (visiting nurses and care managers) should proactively recommend the use of short-stay services to support the health of family caregivers. Among the caregivers, those who work are expected to experience improvements in sleep, especially through respite care.

### 4.4. Study Limitations

This study had a number of limitations. First, participants comprised only 10 family caregivers of people with dementia or individuals providing nighttime care. The detection rate was 36.2%. We calculated the sample size by assuming an LF/HF of 3.360 caregiving days from the previously obtained sleep data of caregivers of ambulatory patients with dementia [14]. However, the actual value was considerably lower (1.722), resulting in a low detection rate in the planned sample size. Considering the burden on caregivers, the study period was set as one night of caregiving and one night of respite care. However, additional long-term data are required to accurately determine sleep status. It is therefore difficult to generalize the results of this study. However, despite the small sample size, the objective data on autonomic nervous activity during sleep are important because they provide further evidence of the value of respite care.

## 5. Conclusions

In this study, we examined the effects of respite care via short-stay services among family caregivers of people with dementia or with nocturnal awakening. On respite days, cardiac sympathetic nervous activity in family caregivers was reduced during the first half of the sleep period. This suggests that regular use of short-stay services can improve sleep status among caregivers, making this an effective form of respite care.

## Figures and Tables

**Table 1 ijerph-17-02428-t001:** Care status information.

*n* = 10		*n* (%) or Mean ± SD (Range)	
Years spent caregiving		5.0 ± 2.9	(1.0–10.0)
Nighttime caregiving (yes)		4	(40.0)
ZBI score		33.9 ± 9.9	(22–46)
Patient characteristics			
Age(y)		83.2 ± 11.8	(57–92)
Sex	Male	4	(40.0)
	Female	6	(60.0)
Relationship to patient			
	Spouse	3	(30.0)
	Own parent	5	(50.0)
	Spouse’s parent	2	(20.0)
Cerebrovascular disease (yes)		2	(20.0)
Dementia (yes)		8	(80.0)
Barthel Index score		46.0 ± 29.3	(0–85)
DBDS score (n = 7)		24.4 ± 12.6	(8–45)
	Patient 15 ^1^ (never)	3	(30.0)
	Patient 16 ^2^ (never)	6	(60.0)

^1^ Patient 15 wakes up at night for no obvious reason. ^2^ Patient 16 wanders in the house at night. Abbreviations: Zarit Burden Interview (ZBI); Dementia Behavior Disturbance Scale (DBDS); standard deviation (SD).

**Table 2 ijerph-17-02428-t002:** Caregivers’ sleep status on caregiving days and respite days.

*n* = 10			Caregiving Day		Respite Day			*p*-Value
Actigraphy								
	Bedtime, 24-hours clock ± min		0:17	(23:12–1:11)	23:35	(23:02–0:24)		0.185
	Wake-up time		6:08	(5:54–6:39)	6:06	(5:50–6:57)		0.646
	Sleep time (min)		378.0	(340.3–481.8)	388.0	(346.0–453.5)		0.386
	Sleep efficiency (%)		94.7	(91.2–97.5)	96.1	(91.6–99.6)		0.508
	Sleep latency (min)		14.0	(3.3–34.5)	12.0	(6.5–21.0)		0.359
	WASO (min)		14.0	(4.5–24.3)	16.5	(1.8–32.8)		0.507
HRV								
	HF amplitude (ms)							
		All sleep	3.172	(2.464–5.880)	3.680	(2.495–4.439)		0.959
		First half of sleep	3.507	(2.260–6.372)	3.880	(2.162–5.262)		0.508
		Second half of sleep	2.869	(1.773–5.514)	3.185	(2.682–3.493)		0.721
	LF/HF							
		All sleep	1.722	(1.410–2.260)	1.567	(1.213–2.311)		0.285
		First half of sleep	1.822	(1.298–2.695)	1.392	(1.015–2.558)		0.013 *
		Second half sleep	1.714	(1.381–2.192)	1.850	(1.549–2.558)		0.721

Value provided are median (25–75th percentile); Abbreviations: heart rate variability (HRV); high frequency (HF); low frequency (LF); * significant difference between groups.

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
