# Peer review of "Effectiveness of Respite Care via Short-Stay Services to Support Sleep in Family Caregivers"

_ijerph, 2020, doi:10.3390/ijerph17072428_

Round 1

Reviewer 1 Report

Introduction is ok.

Methods:

  • More information about the participants, how many participant were contacted and how many returned the application?
  • Why Did you dont calculated a sample?
  • About the actigraph, you described that participants must use heart rate sensor before going to sleep, but there is no reference about actigraph during sleep time.
  • How many days elapsed between caregiving and respite care night? It could affect the results?

Results:

  • Because of the small number of participants included (ten) and a high number of exclusion (seven), I would recommend to perform a comparison between two groups to find differences and to discuss if some of these differences (if there is any) affect the final results.
  • Also, because of the small number of participants, did you perfom an analysis of normality test? if it was appropriate to use a parametric test or non-paremetric test?

Discussion is ok but the methods and results should be reviewed

Reviewer 2 Report

Recover is an important and often neglected topic when it comes to fatigue and sleep deprivation. While focusing on a narrow populations of "workers", the paper exposes the impact of partial, chronic stress and sleep deprivation. 

Up front I felt that the paper would be better positioned by touching a bit on the (limited) recovery literature by Belenky and others. Important in those studies is the idea that it takes multiple days for one to recover from night work/sleep deprivation, so in that context, the findings here are not surprising. It takes time to break routine and begin recovering mentally and physiologically. I am not sure there was enough time here to allow for that, especially as the care givers knew they were going to be back to their role very soon.

I am unfamiliar with the caregiver literature, but I wonder if the fact that they were caring for their own relatives adds an additional or stronger stress component? If there is literature on that, it would be useful to include.

As 60% of the participants were categorized as possibly having a sleep disorder, spending a little time talking about the connection between insufficient sleep, obesity, hormone imbalance and sleep disorders seems worthwhile. Along those lines, was there much napping going on? If so, was it across the board, or by certain participants? Was that sleep time included in total sleep time, and if not, why? Any analysis on caffeine intake? Any differences between those who worked and those who did not?

As you point out, you had a small sample size. Did you calculate statistical power? If so, can you include that information? 

That's about it. I felt that your use of English and the organization of the paper was very good. Thank you for this interesting paper.

Round 2

Reviewer 1 Report

The manuscript can be accepted in the present form